# Crosstalk Between Abiotic and Biotic Stresses Responses and the Role of Chloroplast Retrograde Signaling in the Cross-Tolerance Phenomena in Plants

**DOI:** 10.3390/cells14030176

**Published:** 2025-01-23

**Authors:** Muhammad Kamran, Paweł Burdiak, Stanisław Karpiński

**Affiliations:** Department of Plant Genetics, Breeding and Biotechnology, Institute of Biology, Warsaw University of Life Sciences, Nowoursynowska 159, 02-776 Warsaw, Poland; muhammad_kamran@sggw.edu.pl (M.K.); pawel_burdiak@sggw.edu.pl (P.B.)

**Keywords:** absorbed energy in excess, electrical signaling, reactive oxygen species, phytohormones, hypersensitive response, systemic acquired resistance, acclimation, non-photochemical quenching, cross-tolerance

## Abstract

In the natural environment, plants are simultaneously exposed to multivariable abiotic and biotic stresses. Typical abiotic stresses are changes in temperature, light intensity and quality, water stress (drought, flood), microelements availability, salinity, air pollutants, and others. Biotic stresses are caused by other organisms, such as pathogenic bacteria and viruses or parasites. This review presents the current state-of-the-art knowledge on programmed cell death in the cross-tolerance phenomena and its conditional molecular and physiological regulators, which simultaneously regulate plant acclimation, defense, and developmental responses. It highlights the role of the absorbed energy in excess and its dissipation as heat in the induction of the chloroplast retrograde phytohormonal, electrical, and reactive oxygen species signaling. It also discusses how systemic- and network-acquired acclimation and acquired systemic resistance are mutually regulated and demonstrates the role of non-photochemical quenching and the dissipation of absorbed energy in excess as heat in the cross-tolerance phenomenon. Finally, new evidence that plants evolved one molecular system to regulate cell death, acclimation, and cross-tolerance are presented and discussed.

## 1. Introduction

One of the most significant challenges of the twenty-first century is feeding a growing world population while dealing with an overheated planet and progressive desertification of fertile land. Climate extremes are becoming more common, disrupting agricultural production and posing continuous dangers of starvation [1,2]. It was estimated that more than 800 million people worldwide did not have access to enough food in 2021 [1,3]. Global warming is anticipated to increase the frequency of extreme weather, endangering agricultural crop production [4]. According to calculations, the desert area will increase from 3 to 10% of the total land area [5]. Crucial agricultural crops experience an overall yield loss of about 70% due to adverse conditions in the fields and from the environment. This means only 30% of the yield was produced compared to its yield from genetic potentials [6,7,8,9,10]. Growing demand for crop production and maintaining yield in adverse global warming conditions necessitates vastly expanded plant stress resistance research initiatives.

The frequency and severity of several environmental stresses, such as absorbed energy in excess (AEE), ultraviolet (UV) radiation, severe droughts, and heat waves, increase due to global warming. When taken as a whole, these stressors cause significant losses in agricultural productivity. Stress response phytohormones and signaling molecules, including ethylene (ET), salicylic acid (SA), abscisic acid (ABA), jasmonic acid (JA), and reactive oxygen species (ROS), have been widely described in terms of the regulation of plant stress responses and productivity. During heat and osmotic stress response, these factors and electrical signaling regulate a mechanism known as programmed cell death (PCD), an eventual completion of the cell cycle. Induced PCD in some cells is required for effective stress response, leading to the acclimatization or induction of disease resistance, thus optimizing plant survival and production under stress conditions [11,12,13,14,15,16,17,18,19,20]. Systemic- and network-acquired acclimation (SAA and NAA) processes of chloroplast retrograde signaling happening within and between plants are essential for the induction of PCD [11,12,21,22,23]. It is important to know that stress response to episodes of AEE stress applied to low light acclimated plants can induce disease resistance to the virulent bacterial pathogen and better tolerance to UV-C irradiation episodes due to the induction of the cellular light memory [22,24]. This memory involves molecular, physiological, biochemical, and biophysical changes that last several days after the stress of absorption of energy in excess (AEE) and is specific to the quantity and quality of AEE. Therefore, the current knowledge and understanding of the cross-tolerance phenomena, plant stress responses (SAA, SAR, NAA), and PCD signaling mechanisms is much better than a decade ago. The knowledge on the role of conditional PCD regulators, such as LESION SIMULATING DISEASE 1 (LSD1), ENHANCED DISEASE SUSCEPTIBILITY 1 (EDS1), and CYSTEINE-RICH RLK (RECEPTOR-LIKE PROTEIN KINASE) 5 (CRK5), METACASPASES (MC4 and MC8), SIGNAL RECOGNITION PARTICLES (cpSRP43 and 54), 22 kDa photosystem II protein (PsbS), Ca^2+^ and other ion channels, transcription factors (WRKY, DREB—CBF2 subfamily A-1 of environment RF/AP2), ROS scavenging/generating enzymes, and others involved in the regulation of NAAs, SAAs, and SAR, has been increased recently [11,12,13,14,15,16,20,21,22,23,24]. This review contributes to the further understanding of the systemic- and network-acquired acclimation (SAA and NAA) and the cross-tolerance phenomenon. We present and discuss the role of reactive oxygen species (ROS), the redox status of the plastoquinone pool (PQ), phytohormones, electrical signals (ESs), and non-photochemical quenching (NPQ) waves propagated between cells, leaves, and between different plants in the signaling of cross-tolerance phenomenon. The role of transcription factors (TFs) and other proteins as components of the secondary messenger system activated by these complex waves is also presented. Finally, the holistic picture of induced cross-tolerance that makes the plants capable of conditionally optimizing resilience and yield in multivariable environments is presented.

## 2. Abiotic and Biotic Stresses

In the classical view, environmental stress factors can be classified into two major categories: abiotic and biotic stresses. Plants are constantly exposed to biotic and abiotic stresses, significantly impacting their growth, development, and overall fitness [6,12,25,26,27,28]. Abiotic stresses include factors such as changes in temperature extremes (low or high), light intensity and quality (low light, high light, red/blue/UV ratio), water and osmotic stress, heavy metal toxicity (i.e., Hg, Cd, Pb), and nutrient deficiency (organic/inorganic). In the natural environment, abiotic stresses are usually clustered; for example, a shift from low to high light intensity is associated with a sudden increase in UV radiation and an immediate foliar temperature increase due to absorbed energy in excess and its immediate dissipation as heat via the non-photochemical quenching (NPQ) mechanism [29]. This usually can change plant physiology and biochemistry (metabolism), leading to altered retrograde signaling [21,22,24] and changes in gene expression. As a result of NPQ induction, plants increase foliar heat production from photosystems and inhibit stomatal conductance, leading to impaired foliar cooling (transpiration), decreased CO_2_ assimilation, water and nutrient uptake, and induced photorespiration [17,19,30,31,32,33,34,35,36,37,38,39,40,41,42,43,44,45,46,47]. Mittler et al. (2022) [20] discussed the influence of various abiotic stress factors on plant signaling networks, highlighting the role of ROS and stress-responsive genes in stress perception and adaptation.

Abiotic stresses, such as high light, heat, and drought, often combine and are difficult to separate. AEE alone can induce heat shock response due to the higher dissipation of AEE as heat by non-photochemical quenching (NPQ) [29], and *Arabidopsis thaliana* subjected to a combination of high light and heat stress exhibit a unique metabolic response, including an increased accumulation of sugars and amino acids, as well as decreased levels of metabolites participating in the tricarboxylic acid (TCA) cycle [48]. The combined stress in *Arabidopsis* also leads to the accumulation of non-proteinogenic amino acid γ-aminobutyric acid (GABA), which contributes significantly to plants’ adaptation to intense light and heat stress, possibly via encouraging autophagy [49]. In *Arabidopsis*, the combination of heat and drought stress led to changes in plant metabolomics and root bacterial microbiota, indicating an orchestrated modulation of the whole holobiont [50]. Wheat crops also experience combined heat and drought stress, which has more detrimental effects on growth and yield than individual stresses. Wheat has developed advanced responses at various levels to tolerate these combined stressors [51]. Due to drought and heat stress, cell wall suberization rises in *Arabidopsis* roots, leading to alterations to the biosynthesis and assembly of primary cell wall components [52].

Biotic stresses, arising from interactions with living organisms, pose significant threats to plant health and agricultural productivity [53,54]. These stressors include pathogens, pests, herbivores, and others, which can cause diseases, nutrient deficiency, physical damage, and induce defense responses in plants [55,56,57,58,59,60,61,62]. Plants have evolved intricate mechanisms to detect and respond to pathogens—a two-branched innate immune system. It means recognition and responsiveness to the molecules of microbes, including non-pathogenic ones. Secondly, they react to the pathogen virulence factors directly or through their effects on host targets [63,64,65,66,67]. Plants do not have mobile “killer” cells (e.g., T4 cells) and a somatic adaptive immune system like animals. Instead, they rely on the innate immunity of each cell, mediated by gene pairs (R genes in plants and avr genes in pathogens) that induce programmed cell death in infected and noninfected cells, known as the hypersensitive disease defense response (HR) and systemic acquired resistance [63,64,65,66,67].

An oxidative burst, manifested by rapid ROS generation, is typically present in conjunction with R gene-mediated resistance and is necessary for HR, a PCD believed to restrict pathogen access to nutrients and water. Stimulating SA-dependent signaling pathways that develop particular pathogenesis-related (PR) proteins is also linked to R gene-mediated resistance. Some defensive mechanisms in plants are regulated by systems that rely on ET and/or JA. Different classes of R proteins, such as those with nucleotide-binding (NB) domains and leucine-rich repeats (LRRs), play crucial roles in basal defense regulation and the resistance to various pathogens, showcasing the diversity of plant immune responses [64,68,69,70].

Plant pathogens are frequently classified as either necrotrophs or biotrophs based on their life cycle and how they propagate. While necrotrophs live on dead host tissues, biotrophs need live host tissue to propagate [71]. It is interesting to see how SA signaling and R gene-mediated resistance may lead to resistance to avirulent biotrophs [72]. Such pathogens would have no feeding supply due to the HR response due to PCD induction. However, in the case of necrotrophs, the pathogen would have an easier time existing in the host due to cell death [68]. Studies screening *Arabidopsis* mutants for deficiencies in resistance to different diseases using defective signaling pathways critical in defense gave some evidence to it, like resistance to the necrotrophic fungus *Alternaria brassicicola* is unaffected by the mutation *npr1* and the transgene NahG, which disrupts SA signaling. However, they cause resistance to the biotrophic oomycete *Peronospora parasitica* to be abolished. On the other hand, resistance to the necrotrophic fungus *A. brassicicola* is significantly compromised by the *coi1* mutation, which disrupts JA signaling, whereas resistance to *P. parasitica* is unaffected. These findings showed that plant defensive mechanisms may be modified depending on the pathogen invading, with SA-dependent defenses targeting biotrophs and JA- and ET-dependent responses targeting necrotrophs [68,73,74,75].

## 3. Cross-Tolerance

When plants are exposed to one kind of stress, it can trigger shared signals and pathways, which increases their ability to withstand other types of stress. This process is called cross-tolerance and is associated with increased adaptive fitness [17,21,26,76,77,78]. It can be accomplished by co-activating the plant’s innate immune system, which involves a network of non-specific stress-responsive pathways that bridge biotic–abiotic stress borders. Heat, chilling, drought, and salt stress are frequent abiotic stressors with cross-tolerance effects [79,80]. Different stress signaling networks also interact, which might cause plants to develop cross-tolerance. For instance, *Epichloë* endophyte-infected grasses exhibit more excellent resistance to abiotic stressors like drought through increased root biomass and high stomatal conductance with rich accumulation of solutes for osmotic regulation, to cold by the upregulation of unsaturated fatty acids, to salinity by keeping anatomical changes (like increased xylem, phloem, and vascular bundles size), and to pathogen infections by increasing antioxidants and phenolic compounds that are known to be important for plant defenses [81,82]. Capiati et al. (2006) [83] showed that wounding boosts tomato plants’ resistance to salt through a mechanism involving the systemins and JA. They showed that calmodulin-like activities are necessary for the downstream signaling events that lead to cross-tolerance between wounding and salt stress. Lima et al. (2018) [84] evaluated cross-tolerance induced by heat stress and water stress in common beans and observed improved germination under osmotic stress following heat stress. These findings suggest that agricultural plants may be engineered or bred to resist many abiotic or biotic stresses [15,26,42].

Cross-tolerance involves chloroplast, mitochondrial retrograde, and nuclear anterograde signaling, which are involved in intra- and intercellular communication between chloroplasts and the nucleus. This signaling pathway is mediated by the redox signal, which regulates the expression of the chloroplast- and nuclear-encoded genes required for appropriate stress responses [11,19,20,21,22,23,24,26,42,64,77,78,85,86]. The WHIRLY family of proteins and the REDOX-RESPONSIVE TRANSCRIPTION FACTOR 1 (RRTF1) have been identified as potential mediators of chloroplast-to-nucleus retrograde signaling, leading to cross-tolerance [11,19,20,21,22,23,24,26,42,64,77,78,85,86]. Hydrogen peroxide (H_2_O_2_), a byproduct of various aerobic pathways, has also been implicated in retrograde signaling and cross-tolerance induction [86]. The chloroplast acts as an environmental sensor and communicates with other cell compartments through retrograde signaling to regulate nuclear gene expression in response to developmental cues and stresses by Ca^2+^ and ROS [11,19,20,21,22,23,24,26,85,87,88].

On the other hand, retrograde signaling is also essential for mitochondria biogenesis and stress responses. Nevertheless, further research is required to determine if mitochondria and chloroplasts trigger distinct signaling pathways or if they merge into one [88]. The second option is probable given that the energy, metabolism, and redox state are closely linked to their functioning [89,90,91]. Nonetheless, mitochondrial retrograde communication is believed to be predominant in non-photosynthetic tissues. This theory was supported by the finding that the overexpression of TF ANAC013 in *Arabidopsis* increased the chloroplast’s resistance to photooxidative stress [92,93]. Other reports showed that a strong unfolded protein response (UPR^mt^) is induced by either mitochondrial translation or protein import. In this process, a quick oxidative burst starts the retrograde signal that activates MPK6 and causes a systemic hormone response that is primarily dependent on ethylene signaling but also involves auxin and JA [94]. As a result, an anterograde response is activated, which assists plants under mitochondrial proteotoxic stress conditions by repairing mitochondrial translation through mitochondrial ribosomal proteins (MRPs) and mitochondrial protein import/folding. However, in this review, mostly chloroplast retrograde signaling is discussed.

The mechanism by which plant chloroplasts sense stress stimuli is the quickest process that happens within pico- or nanoseconds (e.g., AEE dissipation as heat and fluorescence, singlet oxygen production, absorbed energy transfer, electron transfer in photosystem II) (Figure 1). Redox reactions in photosystems, such as electrical charge separation, pH gradient imbalances across thylakoid membranes, or the stimulation of the xanthophyll cycle through NPQ, can be triggered by AEE [77,95]. On an interval of microseconds to seconds, ROS deposits can impact hormones, sugars, ion signaling pathways, and the redox state of the PQ and glutathione/ascorbate pools (Figure 1). Specific regulatory proteins, such as CRK5 and MITOGEN-ACTIVATED PROTEIN KINASE 4

(MPK4) regulate stomatal closing within seconds, leading to enhanced photorespiration, triggering ROS generation, and ethylene synthesis [41,96] (Figure 1). This is followed by LSD1-driven regulation, acclimation, and defense reactions and PCD by limiting ROS and ET production [11,17,23,96]. These responses are essential for conditionally optimal plant photosynthesis, transpiration, light acclimation, defense actions, and plant health (Figure 1 and Figure 2) [15,16,17,22,23,24,96,97]. Systemic- and network-acquired acclimation (SAA and NAA) and stress memory serve as “naive” non-acclimated plant cells, chloroplasts, and a PSII training process. Peak et al. (2004) [98] suggested that foliar stomata conductance and chlorophyll fluorescence are regulated by the cellular automaton algorithm that relies on collective dynamics and emergent, distributed biological computation in the leaves of plants that, on the contrary, regulates chlorophyll *a* fluorescence and stomata conductance. Later on, studies suggested that ES-, ROS- and NPQ-wave-like-changes propagated in SAA and NAA could be a process regulated in an identical manner like Peak et al. (2004) suggested [22,23,24,89] (Figure 1 and Figure 2). Plant cells can selectively memorize AEE episodes and the spectral quality of light (either physiologically or molecularly) [22,23,24]. Overall, the interplay between chloroplast retrograde NPQ- and PQ-dependent signaling and other cellular signaling pathways is crucial for the plant’s ability to acclimate and defend, and thus, for cross-tolerance.

Peak et al. (2004) [98] presented confirmation for computation in plans regulating foliar chlorophyll *a* fluorescence and stomatal conductance. In short, under some conditions, stomatal conductance becomes synchronized, and foliar regions with lower chlorophyll *a* fluorescence have higher conductance while regions with higher fluorescence have lower conductance. This regulation can be described by a mathematical cellular automata algorithm [98]. This communication network is controlled and regulated by wavy-like, discrete, and spatially in-time electrical- and ROS-signaling and NPQ changes that depend on the PsbS protein level [22,23,24,95,99], proton motive force, and chlorophyll *a* fluorescence decay time changes [95]. This hardware controls the fate of absorbed energy in photosystems, thus controlling chloroplast retrograde signaling across the whole plant and between plants. As a result, it discreetly and spatially controls stomatal conductance values over time, causing further NPQ wave-like adjustments, thus regulating AEE dissipation as heat [24,95,99,100] (Figure 1).

Górecka et al. (2020) [24] showed that prior induction of the cellular light memory in response to AEE episodes developed cross-tolerance to subsequent UV-C episodes in a PsbS-dependent manner. Photosystem II subunit S (PsbS) is an essential protein in plants that regulates NPQ and thus regulates the balance between absorbed energy dissipated as heat and is required for photochemistry [24,100,101,102]. In this study, the *npq4-1* mutant lacking functional PsbS protein and overexpressing PsbS gene in transgenic *Arabidopsis* line (oePsbS) responded differently to an episode of AEE and in the induction of the cellular light memory and subsequent induction of cross-tolerance to UV-C. After the AEE episode, wild-type and oePsbS but not *npq4-1* plants induced light memory and subsequent cross-tolerance to UV-C. The study revealed a new, significant function for PsbS and NPQ as regulators of chloroplast retrograde signaling and UV cross-tolerance [24].

Systemic acquired acclimation (SAA) is an essential light acclimatory molecular and physiological process that depends on chloroplast retrograde signaling and NPQ [22,23,24,95,99,103]. Ciszak et al. (2015) [95] studied the time-resolved fluorescence of chlorophyll a in the *npq4-1* mutant in leaves exposed to excess light and in leaves undergoing SAA in ambient light conditions. In this study, leaves undergoing SAA in ambient low light and leaves directly exposed to AEE showed the regulation of fluorescence decay (FD). Wild-type *Arabidopsis* leaves exposed to AEE had a significantly shorter FD time than the control leaves and leaves undergoing SAA in ambient low light. However, SAA leaves showed a more minor but significant decrease in FD time compared to the control low-light leaves. These findings suggest that SAA signaling regulates the quantum-molecular properties of the supramolecular complex of photosystem II and that PsbS-dependent light memory processing is necessary for the regulation of SAA [95]. It is important to point out that the synthetic (transgenic) improvement of NPQ and its relaxation in tobacco and soybean lead to an improvement of biomass and seed yield productivity in field conditions [101,102].

Electrical signaling (ES) plays a crucial part in cross-tolerance development. It was reported that when a single dandelion leaf is subjected to punctual wounding or high light stress, it generates a surface foliar electrical signal that then spreads to the nearby in-contact plants [23]. This can cause systemic NPQ and physiological alterations in both plants [23,99]. This electrical signaling can be transmitted from plant to plant in a network of plants connected successively by physical touch. This new physiological phenomenon was called network-acquired acclimation (NAA) [23]. These findings strongly imply that the transmission of direct surface electrical signals between the receiver plant (leaves) and the stressed host plant (or one leaf) results in discrete and spatial changes in the absorbed energy fate (NPQ) as well as the induction of common retrograde signaling for defense responses and acclimatization in the host plant and the entire plant community (Figure 2) [23].

## 4. Signaling Networks During Biotic and Abiotic Stresses

Abiotic factors are more detrimental to crop yield when they occur simultaneously [26,104,105]. It is well known that factors such as drought, heat waves, and salinity affect the development and spread of diseases and insects [104,106,107,108,109,110,111], as well as plant–pest interactions [107]. Many weeds use water more effectively than crops; in that case, abiotic stress, like drought, can favor these interactions [104,109,112]. The cumulative impact of stressors on crops is not necessarily positive since the final result is often determined by the interplay between these factors [113,114,115,116,117] (Table 1).

Crosstalk between abiotic and biotic stresses occurs on physiological and molecular levels. Reduced photosynthesis, decreased water use efficiency (WUE), and changes in stomatal opening are affected by biotic stresses but are essential for a plant’s ability to withstand abiotic stress [118,119,120]. For instance, it was revealed that Na^+^ and Cl^−^ levels in *Phaseolus vulgaris* shoots increased in salinity after being exposed to a root pathogen [118,121]. ABA signaling, which coordinates plant adaptation to abiotic stress, can be diminished by SA signaling, which is generated following invasion with *Pseudomonas syringae* pv. *tomato* [122], whereas ABA synthesis and signaling depend on zeaxantin, a direct precursor of ABA synthesis triggered by NPQ changes in the chloroplast. In another study, HopAM1, a type III effector of *Pseudomonas syringae* that targets HSP70, is involved in the stress-induced closure of stomata in an ABA-dependent manner [123,124]. In *Arabidopsis*, the HopAM1 overexpressed lines enhance the sensitivity toward ABA for stomatal closure and germination arrest [125]. Tolerance to abiotic stress such as drought and freezing can potentially result from plant–microbe interactions, like exposing plants to a viral strain, which includes *tobacco rattle virus* (TRV), *tobacco mosaic virus* (TMV), *cucumber mosaic virus* (CMV), and *brome mosaic virus* (BMV). The infection of these strains is beneficial against drought stress. On the other hand, the CMV strain works in freezing stress in beet crops due to elevated levels of antioxidants and osmoprotectants [118,126,127]. By interacting with hormonal processes, maintaining water and source–sink relationships, and increasing plant vigor under stress, fungi such as fungal endophytes, non-pathogenic rhizobacteria, and mycorrhizal fungi can benefit plants [118,128,129,130].

Research of the impacts of the external use of compounds that activate the plant defense mechanism known as priming provides more evidence for the overlap between abiotic and biotic stress signaling pathways [131]. In *Arabidopsis*, increased resilience to heat, drought, and salt stresses, as well as increased resistance to both biotrophic and necrotrophic fungi, was found when β-aminobutyric acid (β-ABA), SA, and jasmonates were applied [118,132]. Hydrogen peroxide H_2_O_2_ and nitric oxide NO priming for salt tolerance moderately increased the abundance of oxidized and *S*-nitrosylated proteins, which remained relatively similar after applying stress. Non-treated plants were more sensitive and exhibited increased protein carbonylation, oxidation, and higher antioxidant enzyme activity [133,134]. In cauliflower seedlings, the enrichment of H_2_O_2_ and superoxide anion is due to the pre-treatment of H_2_O_2_, which enhances the MDA content. This pre-treatment of H_2_O_2_ produces changes in antioxidant systems, which include enzymatic (SOD, CAT, GPX, and APX) and non-enzymatic systems (AsA, GSH, and proline) [135]. In durum wheat seeds primed with H_2_O_2_, resistance against salinity stress is induced by increasing enzymatic and non-enzymatic antioxidant defense systems [136]. Recently, epigenetic modifications, specifically chromatin-regulated gene activation, have been proposed to govern priming responses [137] (Table 1).

**Table 1 cells-14-00176-t001:** Factors, phytohormones, and chemical compounds that can trigger cross-tolerance in plants.

Factors	Enhanced Resistance Against	Mechanisms	Examples	References
Drought	Biotic Stress (Weeds)	Drought favors plants with efficient water usage, indirectly suppressing weeds’ advantage over crops.	Weeds are more competitive under drought, altering crop–weed dynamics.	[104,109,112]
Salinity (Na^+^ and Cl^−^)	Root Pathogen (*Macrophomina phaseolina*)	Salinity exposure strengthens resistance mechanisms in infected plants.	*Phaseolus vulgaris* exposed to salinity and *M. phaseolina* shows increased Na^+^ and Cl^−^ accumulation in shoots.	[118,121]
Abscisic Acid (ABA)	Pathogen Infection (*Pseudomonas syringae*)	ABA signaling enhances tolerance through stomatal regulation and water-use efficiency.	*P. syringae* triggers ABA-mediated stomatal closure to limit water loss and pathogen entry.	[123,124,125]
SA and Jasmonates	Heat, Drought, and Salt Stress	Defense priming activates stress-related pathways and antioxidant systems.	β-aminobutyric acid (β-ABA), SA, and jasmonates increase tolerance to abiotic stresses.	[118,132]
H_2_O_2_ Priming	Salt Stress	Enhances enzymatic and non-enzymatic antioxidant systems, reducing oxidative damage.	Durum wheat seeds primed with H_2_O_2_ exhibit higher resistance to salinity.	[135,136]
Nitric Oxide (NO) Priming	Salt Stress	Enhances stress-specific protein modification (e.g., S-nitrosylation) and antioxidant enzyme activities.	NO-primed plants show reduced protein oxidation and better salt tolerance.	[133,134]
Plant–Microbe Interactions and Viral Treatment	Drought and Freezing Stress	Fungi, rhizobacteria, and viral strains enhance hormonal interactions and antioxidant, vigor, and osmoprotectant levels.	Non-pathogenic fungi, rhizobacteria, and viral infection increase antioxidant defenses in beet during freezing stress.	[118,126,127,128,129,130]
Epigenetic Modifications	Multiple Abiotic Stresses	Chromatin remodeling creates stress “memory”, enabling faster responses to subsequent stresses.	Epigenetically primed *Arabidopsis* plants show enhanced resistance to heat, drought, and salt stresses.	[137]

### 4.1. Transcription Factors’ Role During Abiotic and Biotic Stress

Because of their functions as key regulators of various stress-related genes, transcription factors (TFs), such as LSD1, EDS1, PAD4, WRKYs, NACs, ERFs, βCAs, DREB, and CBF2, are intriguing prospects for genetic modification [11,130]. The molecular function of LSD1 is currently unknown, but the mutant *lsd1* phenotype is described as runaway cell death (RCD). Runaway cell death (RCD) in LSD1 mutants is usually elicited by AEE, root hypoxia, disturbed stomatal conductance [11,17,138,139], cold [140], drought [11,17], UV radiation [139,141], and pathogen infections [64]. Due to its involvement in different signaling pathways and the regulation of plant CD due to different kinds of stresses (both abiotic and biotic), LSD1 is an important TF in cross-tolerance [11]. Importantly, the RCD of *lsd1* is closely involved in EDS1 and PHYTOALEXIN DEFICIENT 4 (PAD4). These two TFs are important in RCD, as in *eds1*/*lsd1* and *pad4*/*lsd1* mutants, RCD was not found even by applying different plant stresses [21,39,64,77,139]. The ablation of LSD1 and EDS1/PAD4 behave differently toward different stresses, and as a result, the enrichments on ROS and SA are also different [11,17,39,140]. Therefore, LSD1 is the native regulator of EDS1- and PAD4-dependent cellular pathways that lead to CD. LSD1 also greatly impacts cell proliferation and modification processes, especially in non-oxidative stress conditions, which helps plants in growth and development [39,142,143,144,145,146] (Table 2).

Transcription factors like WRKY, NAC, ERF, and βCAs have also been related to improved tolerance in agricultural crops and model plants since they initiate defense-responsive gene expression [147]. Previous studies have shown that MPK3/6 phosphorylated the transcription factors WRKY33 and ERF6 to promote the synthesis of camalexin and activate defense genes [148,149]. In another example, the GhMAP3K15-GhMKK4-GhMPK6, a drought stress-activated MAPK cascade phosphorylated in cotton, can then activate GhWRKY59 to control the plant’s response to the drought stress [150]. The study by Zhao et al. (2021) [56] found that the ZmWRKY104 phosphorylation by ZmMPK6 is important for its role in ABA-induced antioxidant defense and drought tolerance in maize. Another study reported that when H_2_O_2_ levels arise, the ANAC017, located in the ER, releases its N-terminus into the nucleus, which controls primary responses [56]. Recently, it has been shown that ANAC017 contributes to aluminum tolerance by regulating genes that change cell wall synthesis [151]. Along with EDS1 and WRKY33, which react to H_2_O_2_ quickly and transiently, a transcription factor belonging to the ERF family, RRTF1, functions as a key redox signaling regulator [152,153]. In many studies, RRTF1 also showed its role in JA and auxin biosynthesis crosstalk, PCD, salt tolerance, and pathogen resistance [154,155,156,157]. Recent studies by Białas et al. (2024) [158] suggested that the simultaneous overexpression of plant β carbonic anhydrases βCA1 and/or βCA2 with PsbS genes leads to improved photoprotection, acclimation to variable light conditions, and WUE. The analysis of *Arabidopsis* mutants showed that in response to low-temperature stress, CBF genes control 134 genes [159]. Six genes implicated in freezing tolerance were identified by transcriptome analysis as being controlled by DREB and ERF/AP2, including CBF1, ERF105, ZAT6, ZAT12, WRKY33, and WRKY40. These findings reveal a novel chloroplast retrograde regulatory hotspot that relies on bicarbonate absorption, βCAs, and PsbS protein relative levels and shows the importance of TFs in cross-tolerance (Table 2).

**Table 2 cells-14-00176-t002:** Dual role of transcription factors in the regulation of cross-tolerance.

Transcription Factor	Function	Stress Type	Mechanistic Insights	References
LSD1	Conditional negative regulator of cell death, ROS, hormonal homeostasis, photosynthesis, transpiration, and cell wall synthesis.	Abiotic (cold, drought, UV) and biotic (pathogens).	Balances cell division and cell death under oxidative stress. Key regulator in cross-tolerance pathways.	[11,17,39,64,139,141]
EDS1	Positive regulator of the cell death conditionally depends on LSD1. Mediates defense and acclimation gene activation via the triacylglycerol lipase domain.	Biotic and abiotic.	Works closely with LSD1 and PAD4.	[17,19,21,39,142]
PAD4	Promotes disease resistance and abiotic stress responses. Positive regulator of the cell death conditionally depends on EDS1.	Biotic and abiotic.	Works closely with LSD1 and PAD4.	[142,143,144]
WRKYs	Activates defense-related gene expression and antioxidant pathways.	Abiotic and biotic.	WRKY33 and WRKY40 are involved in signaling under H_2_O_2_ and ABA conditions.	[56,148,149,150,153]
NACs	Regulates responses to oxidative stress, aluminum tolerance, and cell wall synthesis.	Abiotic (drought, salt).	ANAC017 responds to H_2_O_2_ and activates genes in the nucleus.	[56,151]
ERFs	A key player in redox signaling, salt tolerance, and pathogen resistance.	Abiotic (salt) and biotic (pathogens).	RRTF1 contributes to JA/auxin crosstalk and PCD.	[152,153,154,155,157]
βCAs	Improves photoprotection, acclimation, and water use efficiency (WUE).	Abiotic (hypoxia, high light stress).	Enhances bicarbonate-driven stress signaling and chloroplast retrograde pathways.	[158]
DREB- CBF2	Controls freezing tolerance and low-temperature stress responses.	Abiotic (cold, hypoxia).	Regulates numerous downstream genes (e.g., CBF1, ERF105).	[158,159]

### 4.2. Reactive Oxygen Species Role During Abiotic and Biotic Stress Condition

Higher plants have developed specialized mechanisms to defend themselves from ROS toxicity and to use ROS as signaling molecules. The chloroplast, mitochondria, peroxisomes, and apoplast are the principal ROS-initiating sites under abiotic stress [19,99,160,161,162,163,164]. If left without control, the ROS level would rise in cells and result in oxidative damage [20]. Antioxidant enzymes, such as superoxide dismutase (SOD), ascorbate peroxidase (APX), catalase (CAT), glutathione peroxidase (GPX), and peroxiredoxin (PRX), represent a major ROS-scavenging force. They are crucial for stress tolerance in plants [165].

Environmental factors, such as high light, UV irradiation, drought, cold, high temperatures, and root hypoxia, can disrupt cellular redox homeostasis, resulting in cell death and limiting the amount of produced biomass [15,19]. It is also widely known that ROS play a role in plants’ response to a pathogen invasion. Hypersensitive response (HR), triggered by the ROS burst generated by NADPH oxidases or apoplastic enzymes, can cause PCD in different cells at the infection sites. This HR mechanism can limit their invasion because of the biotrophic nature of infections (feeding on live cells). In contrast, the surrounding cells develop the capacity to avoid cell death via the propagation of ROS [66]. Callose deposition and cell wall cross-linking, mediated by apoplastic ROS, strengthen the cell wall and prevent pathogens from penetrating it [166]. Plant NADPH oxidases produce ROS in the apoplast, but their accumulation has also been shown in mitochondria, chloroplasts, and even nuclei [167]. Additionally, many abiotic stressors can directly or indirectly activate additional signaling pathways to cause ROS generation in the plasma membrane by the Rboh-signaling pathway, in chloroplasts by the Calvin–Benson pathway, in mitochondria through the ubiquinone pathway, and in peroxisomes by the β-oxidation pathway [168]. Cross-tolerance is achieved through crosstalk between ROS signaling mechanisms and other pathways in activating defense responses [169]. Plants respond to abiotic stress by increasing the accumulation of antioxidant defense systems, including secondary metabolites, such as flavonoids, which scavenge excess ROS and help balance the cellular redox state [170]. The interplay between ROS, redox signals, and antioxidative pathways is important for plant acclimation to stress and cross-tolerance acquisition [171]. Additionally, ROS are involved in organelle-to-organelle and cell-to-cell signaling [172]. Crosstalk between ROS and other signaling molecules, such as reactive nitrogen species (RNS), protein kinases, phytohormones, and secondary messengers like calcium, is important for plant responses to environmental stresses [173] (Figure 1).

### 4.3. Hormonal Response During Abiotic and Biotic Stress

In the control of plant immunological responses, jasmonic acid (JA), salicylic acid (SA), ethylene (ET), and abscisic acid (ABA) are thought to be the key molecules. According to several studies [174,175,176,177], the JA signaling pathway always functions as a significant stress hormone route that frequently communicates with various plant hormones to create an extensive signaling network. Several common elements among JA and other plant hormone signaling pathways have recently been discovered. Jasmonic acid (JA) and SA act antagonistically toward each other but often combine to make plants more resilient to stressors [178]. Recently, processes underlying JA–SA crosstalk have been widely studied. Many genes were reported to be involved in SA–JA antagonism, including MYC2, PDF 1.2 (plant defensin 1.2), TGAs (TF family) [179,180], MAPK, NPR1, ERF1, WRKY62, WRKY70, GRX480 (glutaredoxin 480), ORA59 (octadecanoid-responsive *Arabidopsis* AP2/ERF 59), and JAZs [181]. The discovery of the ortholog NPR1 in the ancestor of all terrestrial plants raises the possibility that JA–SA crosstalk occurs in nearly every plant [182].

According to studies, SA reduces the amount of ET produced due to various abiotic factors. For instance, exogenous SA reduces heat sensitivity in heat-stressed plants by decreasing the ACS activity in wheat and promoting proline metabolism while limiting ET production [1,183]. On the contrary, during ozone (O_3_) stress, ET and SA work together to control cell death in *Arabidopsis* and tobacco leaves [184,185,186]. By increasing the transcription of the genes encoding CHORISMATE MUTASE and PAL, O_3_-induced ethylene enhances SA biosynthesis [186]. The inhibition of either SA or ethylene biosynthesis in *Arabidopsis* reverses the O_3_-induced hypersensitive response phenotype [185]. It is widely known that endogenous SA levels and plant defense mechanisms against biotrophic and hemibiotrophic diseases are positively correlated in plants, as discussed by Glazebrook et al. (2005) [68]. Additionally, the external SA application causes multiple plant species, such as *Fusarium oxysporum*, *Alternaria alternata*, *Magnaporthe grisea*, *Colletotrichum gloeosporides*, and *Xanthomonas* spp., to develop local and systemic acquired resistance against a variety of pathogens [68,187,188,189,190,191,192,193,194,195,196,197].

Jasmonic acid and ethylene play an important, synergistic role in plant resistance to necrotrophic diseases [198,199]. The synergies among ET-stabilized EIN3/EIL1 and JA-activated MYC2 control plant growth and insect resistance. Due to the inhibitory effect of MYC2 on EIN3/EIL1, the transcription of ERF1 (downstream gene of EIN3/EIL1) is suppressed, reducing plant tolerance to necrotrophic fungi [200]. VSP2 and CYP79B3 (wound and herbivore responsive genes, respectively) are suppressed via the JA signaling pathway when EIN3/EIL1 and MYC2 interact; however, on the other side, weakening MYC2’s repression of MYC2 protects against a broad range of herbivores [201,202]. ET and JA can operate oppositely to control several abiotic stress genes, in addition to their combined effects on the functions of EIN3/EIL1, including their targets in rice and *Arabidopsis* [203]. The transcription level of EIN3/EIL1-inducible target genes is decreased in etiolated *Arabidopsis*, which is because the already JA-activated MYC TFs, including MYC2, MYC3, and MYC4), may have actively interacted with EIN3/EIL1 and thus have blocked their ability to bind a DNA target [184,200,203]. The ERF1 is a well-known target of MYC2 and EIN3, modulating responses to various abiotic stressors [204,205,206]. Studies in *Arabidopsis* showed that ERF1 regulates the synthesis of many genes, including JA-, drought-, salinity-, and high-temperature-responsive genes. It has been demonstrated that plants overexpressing ERF1 are more resilient to salt, heat, and drought stress [204,205]. Also, Chen et al. (2021) [184] discussed a JA-triggered reduction in EIN3 activity at the post-transcriptional level. These findings imply that JA and ET can control plant immunity oppositely and that their role is in defense responses.

Abscisic acid is also widely known for its regulatory function in response to abiotic and biotic stresses [184]. Studies suggest that ABA influences several plant functions, including seed dormancy, seed development, abscission, desiccation tolerance enhancement, and, most importantly, stomatal closure [207,208]. Additionally, ABA may be important in situations that are not stressful. Several ABA-dependent secondary messengers can also help plants adapt to biotic and abiotic stress, including nitric oxide (NO), cytosolic free Ca^2+^, and ROS [209,210]. By controlling ABA production and vice versa, several substances, including polyamines (PAs), hydrogen sulfide (H_2_S), and brassinosteroids (BRs), enhance drought tolerance in plants [211,212,213]. Other hormones can also assist the effects of ABA [214,215,216]. A study by Kumar et al. (2016) [217] has reported that ethylene and ABA interact antagonistically, affecting one another’s signaling and biosynthesis pathways. According to genetic evidence in *Arabidopsis*, ethylene production and signaling gene knockouts change a plant’s sensitivity to ABA, which in turn impacts the plant’s ability to withstand abiotic or biotic stresses in several developmental stages [37,216,217,218]. For instance, during seed germination and seedling growth, the *Arabidopsis ACS7* loss-of-function mutant exhibits increased salt tolerance, accumulates ABA content, and is hypersensitive to ABA due to decreased endogenous ethylene levels [37,216] (Figure 1).

## 5. Conclusions

In general, environmental stimuli induce foliar stomata closure and the inhibition of photosynthesis, thus inducing AEE and its dissipation as heat in the NPQ mechanism. AEE alone can simultaneously induce five different types of abiotic stress: heat shock stress, photooxidative stress, photoinhibitory stress, photorespiratory stress, and osmotic stress. This, in turn, activates chloroplast retrograde signaling pathways. Chloroplast and cellular redox potential is regulated by glutathione and ascorbate cycles and by other chloroplast redox potential regulators, for example, ferredoxin-NADP^+^ reductase, thioredoxin, and glutaredoxin. These redox changes induce the synthesis of direct and indirect precursors of phytohormones like ABA, SA, ET, JA, IAA, and other signaling molecules in the chloroplasts of local and directly stressed cells, and in non-directly stressed cells and chloroplasts of systemic cells, tissues, and organs by self-propagating waves of electrical-, ROS-, heat-, and phytohormonal-signaling of SAA and SAR. These signaling waves activate or deactivate various phosphatases, kinase cascades, and transcription factors. As a result, some foliar cells induce PCD due to stomata closure and the induction of the photorespiratory burst of ROS, and other cells induce acclimation and defense responses. All these changes within one plant can be communicated to the other neighboring plants by NAA mechanisms that propagate SAA and presumably SAR within the plant community. Therefore, cross-tolerance between abiotic and biotic stress responses is possible not only within one stressed plant but also within the plant community.

Given a few leaves per plant and hundreds of plants in contact in a meadow, thousands of cells per leaf, several dozen chloroplasts per cell, and thousands of PSII per chloroplast, which are potentially involved in the network of connections, this reveals the complexity of the SAA and NAA signaling and communication network mechanisms that affects trillions of possible communications routes in several m^2^ of a meadow. Plants are environmentally smart and intelligent; they communicate in sophisticated ways globally (within the community of plants), regulate absorbed energy fate, regulate foliar temperature, can count absorbed photons, have quantum-molecular (NPQ-value) memory of AEE episodes, and have physiological (redox) cellular memory; they process this memory differentially at the same time in various foliar cells and tissues to acclimatize and immunize. Thus, plants should be regarded as smart and communicating organisms.

## Figures and Tables

**Figure 1 cells-14-00176-f001:**
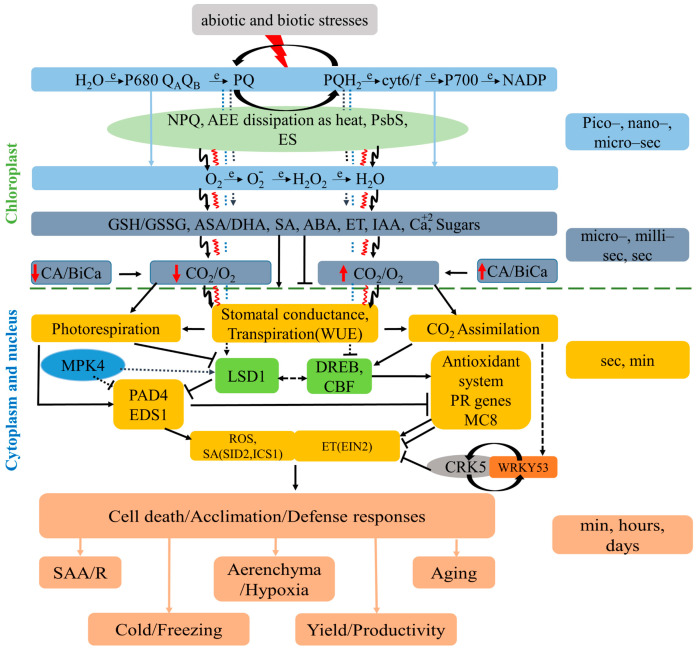
Model of the chloroplast retrograde signaling pathways that regulate cell death (CD) and integrate immune defenses and plant acclimation responses—cross-tolerance to biotic and abiotic stresses. The red lighting-like symbol on the top demonstrates the absorption of energy in excess (AEE) required for electrical charge separation in P680. AEE induces the overproduction of electrons and protons, and thus in non-photochemical quenching (NPQ), heat, and electrical signaling waves. The excess of electrons leads to increased reactive oxygen species (ROS) production and a higher reduction in the plastoquinone (PQ) pool. This triggers redox changes in the stroma in ascorbate, glutathione, and other redox pair component pools, which regulate carbonic anhydrases (CA) activities, and thus, the CO_2_/O_2_ ratio near RuBisCO and sugar production. This, in turn, triggers phytohormone precursor synthesis and photorespiration as a photosynthetic electron sink, which further increases ROS production in peroxisomes and mitochondria and regulates stomatal conductance and programmed cell death (PCD). Chloroplast stroma can be directly connected via stormless with the nucleus, and ES-, ROS-, heat-, and other-signaling molecules or dual transcription factors can be directly transduced to the nucleus and specifically alter nuclear-encoded gene expression and cross-tolerance to abiotic and biotic stresses. This happens in most foliar cells, but some cells induce PCD when the chloroplast outer membrane is disintegrated due to AEE, heat, salicylic acid (SA), and ethylene (ET) signaling. Red wavy-like and black lightning-like arrows show heat and electrical signaling waves, respectively. Solid lines show the confirmed regulatory signaling pathways, while dashed lines show hypothetical signaling pathways. Modified version from Karpiński and colleagues (2014) [11] based on new data [15,16,17,18,19,20,21,22,23,24,41,42,78,89,90,91,92,93,94,95,96].

**Figure 2 cells-14-00176-f002:**
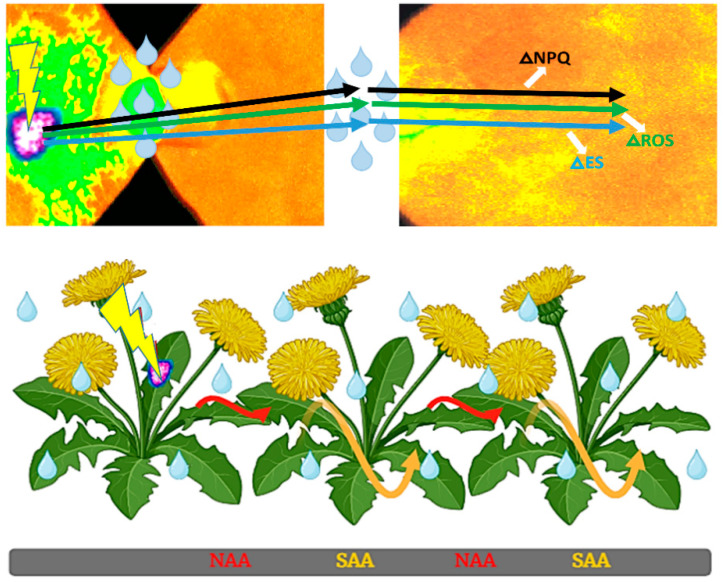
Systemic and network communication of non-photochemical quenching (NPQ) changes between photosystem II and mesophyll cells within and between plants. Electrical signaling (ES) and reactive oxygen species (ROS) waves can induce wavy-like changes in the NPQ value within a chloroplast, between all chloroplasts in the cell, in a whole leaf, in all leaves of a plant, and within the community-dwelling plants, such as the dandelion or *Arabidopsis*, when leaves of neighboring plants are in physical contact [18,19,20,21,22,23,24]. ES and ROS waves transmitted from stressed cells across the whole plant and between plants regulate NPQ- and plastoquinone pool-dependent chloroplast retrograde signaling to induce systemic-acquired acclimation (SAA) (yellow-wavy arrows) in a plant and network-acquired acclimation (NAA) (red-wavy arrows) between plants. Physical contact and electrical conductivity (e.g., a drop of water or high relative humidity) are necessary for NAA. In both transmitter and receiver plants, ES causes spatiotemporal variations in energy quenching (NPQ) (solid black line), the following induction of the ROS wave (solid green line), and electrical signaling (ES) (solid blue line). The yellow lighting-like symbol indicates the stress stimuli. Water droplets show the humidity. Interdependent ion fluxes in the plants cause its variable amplitude. (Model modified from Szechyńska-Hebda et al. (2022) [23]).

## Data Availability

Not applicable.

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
