# Peer review of "Crosstalk Between Abiotic and Biotic Stresses Responses and the Role of Chloroplast Retrograde Signaling in the Cross-Tolerance Phenomena in Plants"

_cells, 2025, doi:10.3390/cells14030176_

Round 1
Reviewer 1 Report
Comments and Suggestions for Authors
This review concentrates on the current understanding of plant stress responses, cross-tolerance to both biotic and abiotic, and on molecular and physiological mechanisms that allow plants to conditionally optimize their growth, development, and seed yield in multi-stress environment. However, the present manuscript should be revised and polished carefully before consideration for publication in Cells.
1. The role of mitochondrial retrograde signaling under abiotic and biotic stresses in plants shoud be included.
(There are many researches in this field,for example, (1)Wang, X., and Auwerx, J. (2017). Systems phytohormone responses to mitochondrial proteotoxic stress. Mol. Cell 68:540-551. e545.
(2)Li J, Yu G, Wang X, Guo C, Wang Y, Wang X. Jasmonic acid plays an important role in mediating retrograde signaling under mitochondrial translational stress to balance plant growth and defense. Plant Commun. 2024 Sep 13:101133. doi: 10.1016/j.xplc.2024.101133. Epub ahead of print. PMID: 39277791.)
2. In the section of 4.3. Hormonal response during abiotic and biotic stress, abscisic acid (ABA) should be considered and included.
Author Response
Subject: Response to Reviewers’ Comments for Manuscript ID: cells-3368017
Dear Reviewer,
We would like to express our gratitude to the reviewers for their insightful and constructive comments on our manuscript, titled “Crosstalk between abiotic and biotic stress responses and the role of chloroplast retrograde signaling in the cross-tolerance phenomena in plants”. We have carefully addressed all the suggestions and made the necessary revisions to enhance the quality of our manuscript. Below, we provide a detailed point-by-point response to the reviewer 1 comments.
Response to the Reviewer 1 comments
Comment 1: The role of mitochondrial retrograde signaling under abiotic and biotic stresses in plants should be included.
Response: We appreciate this valuable suggestion. A paragraph discussing mitochondrial retrograde signaling under abiotic and biotic stress has been added to the manuscript (Lines 188–203). This section references the suggested studies (Wang & Auwerx, 2017; Li et al., 2024 and also other research articles) to provide better understanding of this topic.
Comment 2: In the section of 4.3, hormonal response during abiotic and biotic stress, abscisic acid (ABA) should be considered and included.
Response: As suggested, we have included a detailed discussion of ABA hormonal signaling in Section 4.3. This addition spans Lines 5518–535.
We hope the revised manuscript meets the Reviewer 1 and the journal's expectations. Please let us know if further modifications are required.
Thank you for considering our submission.
Best regards,
Prof. Stanislaw Karpinski
(on behalf of all the authors)
Reviewer 2 Report
Comments and Suggestions for Authors
The review paper contains a lot of useful information.
The text is complex, and at times it is difficult to read because of mistakes in the verbs or in the construction of sentences. I think a re-reading by a professional editor could improve readability.
Just because the text is so complex, and particularly full of abbreviations, it is almost impossible to get a clear picture of the processes described. The two figures are helpful but they are cited only in few instances. Much of the information provided is just in text format, and there is no effort to construct schemes or use tables to critically discuss the findings. The reader does not get a clearer picture of the complex processes, at the end.
When performing reviews, it is common to report the names of plants, fungi, bacteria as they were mentioned in the original research papers. However, it often happens that the scientific binomial names have been changed in the meantime. Therefore, authors should recehck in updated databases that they are using the correct currently accepted names, and report the authority for the name at first mention- for future readers.
The legend to Figure 2 does not refer to what is shown in the pictures making it difficult to understand.
L501 and L553 the citation to Figure 2 seems inappropriate
L591-592 I would take out this sentence.
Comments on the Quality of English LanguageThe text is complex, and at times it is difficult to read because of mistakes in the verbs or in the construction of sentences. I think a re-reading by a professional editor could improve readability.
Author Response
Subject: Response to Reviewers’ Comments for Manuscript ID: cells-3368017
Dear Reviewer,
We would like to express our gratitude to the reviewers for their insightful and constructive comments on our manuscript, titled “Crosstalk between abiotic and biotic stress responses and the role of chloroplast retrograde signaling in the cross-tolerance phenomena in plants”. We have carefully addressed all the suggestions and made the necessary revisions to enhance the quality of our manuscript. Below, we provide a detailed point-by-point response to the reviewer comments.
Response to the Reviewer 2 comments
Comment 1: The text is complex, and at times it is difficult to read because of mistakes in the verbs or in the construction of sentences. I think a re-reading by a professional editor could improve readability.
Response: The manuscript has been thoroughly polished for grammar, sentence construction, and readability. We believe the text is now clearer and more concise.
Comment 2: Just because the text is so complex, and particularly full of abbreviations, it is almost impossible to get a clear picture of the processes described. The two figures are helpful but they are cited only in few instances. Much of the information provided is just in text format, and there is no effort to construct schemes or use tables to critically discuss the findings. The reader does not get a clearer picture of the complex processes, at the end.
Response: To improve readability, two tables have been added to provide a critical discussion of the findings. These tables summarize key processes and transcriptional factors functions in cross-tolarance, ensuring the information is accessible and easier to comprehend.
Comment 3: When performing reviews, it is common to report the names of plants, fungi, bacteria as they were mentioned in the original research papers. However, it often happens that the scientific binomial names have been changed in the meantime. Therefore, authors should recehck in updated databases that they are using the correct currently accepted names, and report the authority for the name at first mention- for future readers.
Response: We have verified and updated all scientific names in the manuscript, ensuring they align with the latest binomial nomenclature standards.
Comment 4: The legend to Figure 2 does not refer to the content shown, making it difficult to understand.
Response: The legend of Figure 2 has been modified to align with its content, making it clearer for readers.
Comment 5: Citations to Figure 2 on Lines 501 and 553 seem inappropriate.
Response: These citations have been corrected. The references now appropriately cite Figure 1 instead of Figure 2.
Comment 6: The sentence on Lines 591–592 should be removed.
Response: The suggested sentence has been removed from the manuscript.
We hope the revised manuscript meets the Rreviewer’s 2 and the journal's expectations. Please let us know if further modifications are required.
Thank you for considering our submission.
Best regards,
Prof. Stanislaw Karpinski
(on behalf of all the authors)
Reviewer 3 Report
Comments and Suggestions for Authors
Dear Authors
Regarding to the manuscript ID: cells-3368017
Title: Crosstalk between abiotic and biotic stresses responses and the role of chloroplast retrograde signaling in the cross-tolerance phenomena in plants
The manuscript is generally is need improvement, the topic is original and relevant in the field of plant stress.
To improve the manuscript, there several points need to be addressed:
- Some sentences are lengthy and contain multiple ideas or clauses, making them difficult to understand.
- Breaking them down into shorter, clearer sentences would enhance readability. Additionally, there are instances of awkward phrasing and incomplete sentences that could be addressed.
- The manuscript could be better organized to present the information in a more logical and coherent manner.
- The flow between sections could be smoother, providing a clearer narrative for readers. The introduction could provide a stronger background and context for the study, leading to the research objectives and hypotheses.
- The manuscript should be focusing on the most important signals of the biotic and abiotic stress related to chloroplast instead of taking about too much signals without deep discussion and understandable mechanisms of action.
- It should also highlight the significance of the research
- The sentences should not be started with abbreviation; it is better to start with complete form, please correct all over the manuscript.
- In general, the use of personal pronouns such as "we, our; i, etc." is not recommended in scientific language. Please revise the manuscript and avoid using it as much as it is possible. Please revise and correct all over the manuscript.
- -The references should be revised carefully in the references list as well as in the text and rewrite according the journal format and the title of the cited articles with an uppercase letter in the initial only for its first word.
- There are some writing mistakes, so the manuscript should be revised carefully.
- There are several comments provided in the attached manuscript should be taken in consideration.
Best regarding

The English must be improved to be more clearly express the research.
Author Response
Subject: Response to Reviewers’ Comments for Manuscript ID: cells-3368017
Dear Reviewer,
We would like to express our gratitude to the reviewers for their insightful and constructive comments on our manuscript, titled “Crosstalk between abiotic and biotic stress responses and the role of chloroplast retrograde signaling in the cross-tolerance phenomena in plants”. We have carefully addressed all the suggestions and made the necessary revisions to enhance the quality of our manuscript. Below, we provide a detailed point-by-point response to the reviewer comments.
Response to the Reviewer 3 comments
Comment 1: Some sentences are lengthy and contain multiple ideas or clauses, making them difficult to understand. Breaking them down into shorter, clearer sentences would enhance readability. Additionally, there are instances of awkward phrasing and incomplete sentences that could be addressed.
Response: We have revised all runaway sentences and shortened them, improving clarity and readability throughout the manuscript.
Comment 2: The manuscript could be better organized to present the information in a more logical and coherent manner.
Response: The manuscript has been reorganized to ensure a logical and coherent flow of information.
Comment 3: The flow between sections could be smoother, providing a clearer narrative for readers. The introduction could provide a stronger background and context for the study, leading to the research objectives and hypotheses.
Response: The flow between sections has been improved, and the introduction has been revised to provide a stronger background and highlight the objectives and hypotheses.
Comment 4: The manuscript should be focusing on the most important signals of the biotic and abiotic stress related to chloroplast instead of taking about too much signals without deep discussion and understandable mechanisms of action.
Response: Unnecessary details regarding other signaling pathways have been removed to focus on chloroplast signaling and its role in stress responses.
Comment 5: It should also highlight the significance of the research
Response: The significance of our research has been highlighted in a dedicated paragraph (Lines 69–78).
Comment 6: The sentences should not be started with abbreviation; it is better to start with complete form, please correct all over the manuscript.
Response: Sentences starting with abbreviations have been corrected throughout the manuscript.
Comment 7: In general, the use of personal pronouns such as "we, our; i, etc." is not recommended in scientific language. Please revise the manuscript and avoid using it as much as it is possible. Please revise and correct all over the manuscript.
Response: All personal pronouns have been removed, and the text has been revised for appropriate scientific language.
Comment 8: The references should be revised carefully in the references list as well as in the text and rewrite according the journal format and the title of the cited articles with an uppercase letter in the initial only for its first word.
Response: The references have been carefully revised to conform to the journal’s formatting requirements.
Comment 9: There are some writing mistakes, so the manuscript should be revised carefully.
Response: The manuscript has been thoroughly reviewed and revised to eliminate grammatical and writing mistakes.
Comment 10: There are several comments provided in the attached manuscript should be taken in consideration.
Response: All comments provided in the annotated manuscript have been addressed and incorporated into the revised version.
We hope the revised manuscript meets the Reviewers' 3 and the journal's expectations. Please let us know if further modifications are required.
Thank you for considering our submission.
Best regards,
Prof. Stanislaw Karpinski
(on behalf of all the authors)
Round 2
Reviewer 1 Report
Comments and Suggestions for Authors
The authors revised the manuscript according to my suggestions and concerns.
Reviewer 2 Report
Comments and Suggestions for Authors
The paper has been significantly improved
Reviewer 3 Report
Comments and Suggestions for Authors
Dear Authors
I appreciate your concern to the required comments
Comments on the Quality of English LanguageThe English used is correct and readable